# Antimicrobial Activity of Biogenic Silver Nanoparticles from *Syzygium aromaticum* against the Five Most Common Microorganisms in the Oral Cavity

**DOI:** 10.3390/antibiotics11070834

**Published:** 2022-06-21

**Authors:** Erika Alejandra Jardón-Romero, Edith Lara-Carrillo, María G. González-Pedroza, Víctor Sánchez-Mendieta, Elías Nahum Salmerón-Valdés, Víctor Hugo Toral-Rizo, Oscar F. Olea-Mejía, Saraí López-González, Raúl A. Morales-Luckie

**Affiliations:** 1Center for Advanced Studies and Research on Dentistry, Autonomous University of the State of Mexico (UAEMex), Toluca 50200, Mexico; erika_abisma@hotmail.com (E.A.J.-R.); elarac@uaemex.mx (E.L.-C.); sarailogo@hotmail.com (S.L.-G.); 2Faculty of Sciences, Department of Biotechnology, Autonomous University of the State of Mexico (UAEMex), Toluca 50200, Mexico; mggonzalezp@uaemex.mx; 3Joint Center for Research in Sustainable Chemistry (CCIQS), Department of Material Science, Autonomous University of the State of Mexico (UAEMex), Toluca 50200, Mexico; vsanchezm@uaemex.mx (V.S.-M.); ofoleam@uaemex.mx (O.F.O.-M.); 4School of Dentistry, Autonomous University of the State of Mexico (UAEMex), Toluca 50130, Mexico; ensalmeronv@uaemex.mx (E.N.S.-V.); vhtoralr@uaemex.mx (V.H.T.-R.)

**Keywords:** biosynthesis, silver nanoparticles, *Syzygium aromaticum*, oral microorganisms

## Abstract

*Syzygium aromaticum* (clove) has been used as a dental analgesic, an anesthetic, and a bioreducing and capping agent in the formation of metallic nanoparticles. The main objective of this study was to evaluate the antimicrobial effect in oral microorganisms of biogenic silver nanoparticles (AgNPs) formed with aqueous extract of clove through an ecofriendly method “green synthesis”. The obtained AgNPs were characterized by UV-Vis (ultraviolet-visible spectroscopy), SEM-EDS (scanning electron microscopy–energy dispersive X-ray spectroscopy), TEM (transmission electron microscopy), and ζ potential, while its antimicrobial effect was corroborated against oral Gram-positive and Gram-negative microorganisms, as well as yeast that is commonly present in the oral cavity. The AgNPs showed absorption at 400–500 nm in the UV-Vis spectrum, had an average size of 4–16 nm as observed by the high-resolution transmission electron microscopy (HR-TEM), and were of a crystalline nature and quasi-spherical form. The antimicrobial susceptibility test showed inhibition zones of 2–4 mm in diameter. Our results suggest that AgNPs synthesized with clove can be used as effective growth inhibitors in several oral microorganisms.

## 1. Introduction

The ideal properties of an antibacterial coating include prolonged activity, high levels of bactericidal and bacteriostatic activity, ability to act against a wide spectrum of bacteria, biocompatibility, and low in vivo toxicity [1,2,3,4,5]. Historically, silver compounds and ions have been extensively used for hygienic and healing purposes. However, over time, their application as an anti-infection agent has dwindled due to the advent of antibiotics and other disinfectants [2,3,6,7]. Recently, there has been renewed interest in manufactured silver nanomaterials, thanks to their unusually strong physicochemical properties and biological activities compared to their bulk parent materials [1,4]. Today, it should be noted that nanomaterials have managed to enter different regulatory and safety fields (such as clinical trials and good manufacturing practices) [6], this to regulate and guarantee the quality of different areas of management and production [7]. In addition, silver nanoparticles (AgNPs) synthesized by green methods have many other applications in different biotechnological areas such as water filtration agents [8], disinfection and preservation of foods [9,10] and various materials [11], the production of cosmetics [10,12], nanoinsecticides and nanopesticides [13], nanocomposites [14], amongst others [15,16,17,18,19,20]. Depending on the technique used, their synthesis can be divided into the following: chemical methods involving the reduction or precipitation of metals in the presence of stabilizing agents; physical methods such as thermolysis, photochemical, and sonochemistry; finally, biological methods [16,21,22,23,24,25]. Biological methods are extremely important, since reducing agents of a chemical nature are not required. In biological methods, reducing agents are obtained from compounds present in natural extracts. This is the case of the synthesis being used in this research, which also has great advantages, because it is an easy and low-cost method [26]. Compared to chemical methods, biological methods represent less toxicity and are more respectful of the environment, since the most used reducing agents at the industrial level and within chemical methods are sodium borohydride, hydrazine and hypophosphite, which can increase environmental toxicity or biological hazards. In addition, capping agents, such as polyvinyl alcohol, must be used to prevent the AgNPs from aggregating. Another issue is that the high temperature may also increase the production cost [16,27]. Fortunately, the biological synthesis of nanoparticles (NPs), also known as “green synthesis,” has allowed the formation of metallic nanostructures from the use of bacteria, fungi, plants, or their extracts, meaning that this approach to synthesis is a non-toxic and environmentally friendly alternative. Sometimes, the deployment of this synthesis equals or exceeds the expectations of NPs synthesized by physical and chemical methods, in terms of cost and characteristics, as previously described [27,28,29,30]. Green extracts contain molecules that carry hydroxyl moieties in their functional groups, mainly of the phenolic type, which can be used for the reduction of metal ions and formation of stable complexes with metallic NPs [1,31,32]. AgNPs show efficient antimicrobial properties compared to other metallic NPs, due to their large surface area, which provides better contact with microorganisms. Although AgNPs have been reported to be involved in a wide range of molecular processes within microorganisms, the mechanism of action is still being studied [33], It is important to note that there are not only bacteria that cause conditions in the oral cavity, but also some yeasts such as *Candida albicans* [34], Recently, it has been shown that AgNPs induce alterations in fungal cells and the formation of pores on the cell surface, in addition to changes in membrane fluidity, all of which may be related to changes in the lipid constitution of the plasma membrane and membrane depolarization [35]. Regarding the mechanism of action of AgNPs on bacteria, it has been studied that they have the ability to anchor and subsequently penetrate the bacterial cell wall, which causes structural changes and cell death [36]. The formation of free radicals by the AgNPs may be considered another mechanism by which cells die. Diverse studies, in which electron spin resonance spectroscopy was used, suggest that free radicals form when the NPs come into contact with the bacteria [37]; these radicals are able to damage the cell membrane, rendering it porous, which can ultimately lead to death [2]. It is also true that the interaction of the AgNPs with the sulfur and phosphorus major components of DNA can lead to problems in the DNA’s replication of bacteria (such as epigenetic changes) [28]. Moreover, when it comes to microbial flora, the oral cavity is one of the most densely populated sites of the human body [38]. Over 700 bacterial species or phylotypes have been detected in this location, of which more than half have not been cultivated [39]. This means that microorganisms of the oral community should display extensive interactions when forming biofilm structures, carrying out physiological functions, and inducing pathogenesis [40,41]. The oral cavity is a complex ecosystem that is inhabited by more than 300 bacterial species. Some of these have been implicated in oral diseases such as caries and periodontitis, which are among the most common bacterial infections in humans [42]. The bacteria colonize the teeth in a reasonably predictable sequence. The first or primary colonizers tend to be aerobic (especially *streptococci*, which constitute 47–85% of the cultivable cells found during the first four hours after professional tooth cleaning); as plaque oxygen levels fall, the proportions of Gram-negative microorganisms tend to increase [43]. In the normal oral cavity, several species of the genera *Streptococcus*, *Lactobacillus*, *Lactococcus*, *Enterococcus*, *Staphylococcus*, *Corynebacterium*, *Veillonella* and Bacteroides stand out for being responsible for various oral conditions [44]. Silver compounds and NPs have already been used as dental restorative material, endodontic retrofill cements, dental implants, and caries inhibitory solutions. Despite the effectiveness shown by AgNPs in dental practice, controversy remains over their toxicity in biological and ecological systems, due to the cytotoxicity caused by high concentrations or the specific size of the nanoparticles [45,46]. *Syzygium aromaticum* (Clove) is employed in Indian ayurvedic medicine, Chinese medicine, and western herbalism, while in dentistry, its essential oil is used as an anodyne (painkiller) for emergencies. Clove oil is a pale-yellow liquid with a characteristic odor and taste. It consists of 81–95% phenols (eugenol with about 3% of acetyl eugenol), sesquiterpenes (a- and b-calyophyllenes), and small quantities of esters, alcohols, and ketones [24,27,28,29,30,36,41,45,46,47,48,49,50,51,52]. The high level of eugenol present in clove oil endows it with intense biological and antimicrobial activity [53], which is why it was explored here as a stabilizing and reducing agent in the synthesis of AgNPs. Therefore, it is important to develop dental materials with antibacterial activity and better mechanical properties, which could be manufactured and employed in future clinical applications.

Herein, AgNPs were synthesized using aqueous clove extract, as a reducing and stabilizing agent, taking advantage of the AgNPs antimicrobial properties in synergy with the clove dental applications as an analgesic and anesthetic. The green-synthesized AgNPs were characterized by ultraviolet-visible (UV-Vis) spectrometry, scanning electron microscopy–energy dispersive X-ray spectroscopy (SEM-EDS), transmission electron microscopy (TEM) and ζ potential. The antibacterial activity of the AgNPs against Gram-positive and Gram-negative microorganisms and yeast were tested by the disc-diffusion method.

## 2. Results

### 2.1. Synthesis of Silver Nanoparticles

The reduction of silver ions and formation of AgNPs occurred after silver nitrate solution was placed in contact with the clove aqueous extract, followed immediately by a change in color of the suspension, from colorless to yellow. This process was carried out under ambient conditions. It is relevant to mention that no stabilizing or capping agent was used.

### 2.2. Characterization of Synthesized Silver Nanoparticles

#### 2.2.1. UV-Vis Spectroscopy

The spectra were recorded when both the color and absorption intensity of the colloidal samples remained constant, as shown in Figure 1. Each surface plasmon resonance (SPR) maximum was ubicated in the range of 431–447 nm. In addition, no surface plasmon resonance was observed at more than 500 nm, indicating that most of the AgNPs obtained were of small size and similar shape. As time increases, AgNPs stabilize and reach their maximum growth.

#### 2.2.2. SEM-EDS Analysis

The characterization technique was consistent with the EDS analysis. AgNPs generally showed absorption peaks at approximately 3 keV (Figure 2).

#### 2.2.3. TEM

TEM was employed to characterize the size, shape, and morphology of the synthesized AgNPs, which resulted to be quasi-spherical, with sizes ranging 4–16 nm, with a mean size of 10 nm and a standard deviation of 4.27 nm (Figure 3).

#### 2.2.4. ζ Potential

The electrostatic stabilization of the AgNPs was estimated by measuring their ζ potential values, which were found to be in the range of −15.7 to −16.2 mV.

### 2.3. Antimicrobial Activity

#### 2.3.1. Disk Diffusion Test

The biogenic AgNPs were tested for antimicrobial activities against Gram-positive, Gram-negative bacteria and yeast, presenting similar inhibition zones in all cultures. The inhibition zones produced by the AgNPs displayed halos of 2–4 mm in diameter. The control group (clove extract) did not show any antibacterial effect only antifungal effect (Figure 4). This last datum provides information regarding the efficacy provided by the clove extract only against *Candida albicans*, which turns out to be really important. In Figure 4E, it is easy to perceive this important antifungal activity, and it is observed to be slightly enhanced by the AgNPs. Regarding the results obtained in bacteria, as previously described, it is possible to see the activity of the potentiated extract.

#### 2.3.2. Microdilution in Broth Test

Through this study, we were able to determine both the maximum effective concentration (MEC), as well as the minimum inhibitory concentration (MIC), where we can observe the effectiveness of the NPs at different concentrations against the different microorganisms mentioned above (Figure 5a and Table 1). Specifically, in Figure 5a, it is possible to observe the complete inhibition at the highest concentration; likewise, it is reflected in Figure 5b how Microbial growth after exposure to determined concentrations of synthesized silver nanoparticles. This information is reflected in Table 1. A very important result is that of the activity it has against the bacteria *E. coli*, where the antibacterial activity is completely inhibited until the last treatment.

Table 1 shows, for the five types of microorganisms, the minimum inhibitory concentration and the concentrations in milligrams/mL, as well as the maximum effective concentration of the AgNPs.

## 3. Discussion

Clove is one of the most valuable of all spices, and it has been used for centuries as a food preservative and for medicinal purposes, as an analgesic and antispasmodic, with eugenol being the main compound responsible for this activity. This plant represents one of the richest sources of phenolic compounds such as eugenol, eugenol acetate, and gallic acid (75–77%) [54,55,56,57]. Several authors [51,54,56] have studied the antimicrobial effects of clove extract. For example, Duhan used various plant extracts with high concentration of eugenol, including clove, and compared them with 3% sodium hypochlorite against *Enterococcus faecalis*. Meanwhile, Mansourian and Rana used clove extract against *C. albicans* and compared it to nystatin. In both cases, the results obtained with the clove extracts were more effective than the substances they were compared to. The most probable pathway for AgNPs biosynthesis is that the flavonoids, from the clove buds, act as reducing agents when they encounter Ag^+^ ions. The electrons are transferred and the reduction process occurs, leading to the formation of AgNPs. It is well known that flavonoids also prevent agglomeration and stabilize the AgNPs in aqueous solution [16,21,22,23,58]. This makes a strong case for the involvement of flavonoids in rapid biosynthesis and for the stability of metallic NPs in the aqueous medium [17,59]. Upon storage of the bio-functionalized AgNPs that were synthesized using macerated clove solution, it was observed that the colloidal solution maintained its stability and uniformity. The biosynthesis of AgNPs by clove extract is carried out rapidly, the solution changes color as a function of time, the kinetics of formation (Figure 1) indicates that the reaction stabilizes after 6 h. The process of biosynthesis in this study was carried out under ambient conditions. Qian Sun [60], Deshpande [49], and Bajpai [55] performed different studies to generate AgNPs using a similar green synthesis approach, obtaining antibacterial properties such as those in our study under the same conditions. In addition, Jeevika [48] produced silver nanowires via biosynthesis using clove oil. It is well known that AgNPs show a yellowish-brown color in aqueous solution; this color is a result of the excitation of surface plasmon vibrations in the metal NPs [48,49].

According to UV-Vis spectroscopy results, in the first 20 min of reaction, there was an incipient reduction in which the SPR could already be observed. This may be attributed to the high availability of reducing agents such as eugenol in the aqueous extract. During the next six hours of reaction, the SPR was clearly observed at 447 nm, indicating the formation of AgNPs. After six hours of reaction, SPR with a defined shape appears at 447 nm, indicating that the NPs now have a defined shape and size, compared to plasmons at 431 nm at short reaction times, indicating incipient nanoparticle formation. Measurements were made approximately once every five minutes to verify the AgNPs kinetics of formation.

EDS analysis of the AgNPs confirmed the presence of metallic Ag (see Figure 2), according to [61,62]. It is important to point out that these biogenic AgNPs exhibited certain low size polydispersity and similar shapes, making our method distinct from other bio-reductors; this was corroborated by the TEM micrographs, whereby the image in Figure 3A showed that AgNPs prepared using clove extract have a considerably low polydispersity and an almost spherical morphology. The particle size distribution histogram was determined using TEM images. AgNPs diameters range between 4 and 16 nm, with a standard deviation of 4.2 nm (Figure 3A).

The HR-TEM image in Figure 3B shows AgNPs of 8–10 nm in diameter. This image possesses atomic resolution; therefore, several crystalline planes are distinguishable and the interplanar distances can be measured. The interplanar distances shown in the micrograph of these NPs, wherein the corresponding crystalline planes were specified. The interplanar distances and their corresponding crystalline planes match those of metallic Ag (face-centered cubic structure). The measured interplanar distance was 2.4 Å, which corresponds to the plane [111]. Generally, a suspension that exhibits an absolute zeta potential of 0–100 mV is ubicated on the threshold of agglomeration, and the stability of the particles from the solution will be increased with higher values of approximately −100 mV [63,64]. In this study, the zeta potential of AgNPs was in the range of −15.7 to −16 mV. These values allow us to corroborate an acceptable stability of the AgNPs colloidal suspension.

The microorganisms selected in this study for microbiological tests are part of the normal and pathogenic microflora of the oral cavity. *Streptococcus mutans*, the main causal agent of tooth decay and periodontitis, is found in 70–90% of the population. Meanwhile, *Staphylococcus aureus* is found in patients with bacterial endocarditis and cellulite. *Enterococcus faecalis* is present in most failed root canal treatments. *Escherichia coli* has been isolated from salivary gland infections, while *Candida albicans* is found in one in every 1000 patients attending dental consultation and also in stomatitis associated with the use of dental prostheses. Taken together, these bacteria are responsible for various opportunistic infections in immunocompromised patients. In this study, the antibacterial activity of AgNPs against four types of bacteria and one yeast was investigated. The inhibition zones were similar in all cell cultures. Biological tests of AgNPs against test strains showed that they have a significant effect on the growth of Gram-positive and Gram-negative bacteria. The latter have a layer of lipopolysaccharides at the exterior, while underneath lies a thin layer of peptidoglycan. Thus, a potent antimicrobial effect can be determined in the microdilution in broth study that emphasizes the antibacterial and antifungal effectiveness of the AgNPs as shown in Figure 5b, taking into account the efficacy in *Escherichia coli* having microbial effect until the last concentration of NPs. On the other hand, we observed bacterial growth of the *Streptococcus mutans* strain from the third highest concentration, and also the following *Staphylococcus aureus* and *Enterococcus faecalis*; on the other hand, we see that the antifungal effectiveness of the AgNPs against *Candida albicans* is low but not null. Several studies indicate [1,30,65,66] that the antibacterial activity of AgNPs is based on the NPs attachment to the bacterial cell wall, or the formation of free radicals. In addition, the silver ions released from AgNPs may play a vital role in the antibacterial activity, due to their interaction with the thiol groups of enzymes. Although the lipopolysaccharides are composed of covalently linked lipids and polysaccharides, they have a lack of strength and rigidity. The negative charges on them are attracted toward the weak positive charge available on AgNPs [67,68]. This indicates that AgNPs have great potential to be used in biomedical applications.

## 4. Materials and Methods

### 4.1. Synthesis of AgNPs

In this experimental study, the chemical precursor used was silver nitrate (AgNO_3_) (Sigma-Aldrich, St. Louis, MO, USA). The AgNPs were directly synthesized with the clove (Terana, Especias selectas, Ciudad de México, México) extract using a simple green synthesis procedure. Initially, various concentrations were tested until the optimum conditions were established.

A 10 mM solution of AgNO_3_ was prepared using deionized water. To prepare the reducing agent, 1 g of chopped clove was added to 100 mL of boiling deionized water for five minutes. The mixture was allowed to cool before being filtered into a vacuum flask using a Buchner funnel and Whatman filter paper no. 5. The aqueous extract was mixed, at room temperature, with AgNO_3_ in 3:1 ratio. The resultant mixture was kept undisturbed in a dark place. After a couple of hours, the color of the solution changed due to the formation of AgNPs. The process of biosynthesis was carried out under ambient environmental conditions (that is, at room temperature and under atmospheric pressure); the reaction was completed within a few minutes.

### 4.2. Characterization of Synthesized Silver Nanoparticles

#### 4.2.1. Spectroscopy UV-Vis

The UV-Vis analysis was performed using a spectrophotometer (CARY 5000 Conc UV-Vis spectrophotometer, Varian, Inc., Palo Alto, CA, USA), which was operated at a resolution of 1 nm at room temperature. The spectral range was 300–800 nm. In this way, the kinetics of the reduction were followed until stable NPs were obtained.

#### 4.2.2. SEM-EDS

The final product was sonicated for 30 min to break up larger nanoparticle agglomerates; then, the particles were dried in a vacuum at room temperature (20 °C) prior to analysis. The NPs were attached to aluminum stubs with conductive tape, coated with carbon, and observed under SEM (JEOL, JSM-6510LV at 20 kcV, Tokyo, Japan) with secondary electrons at ×100, ×500, and ×3000 magnification that was operating at 20 kV. EDS analysis was developed.

#### 4.2.3. TEM

TEM was obtained via a JEOL JEM-2100 microscope (Tokyo, Japan). Samples for the TEM examination were prepared by placing a drop of the suspension on a copper grid (300 mesh) coated with carbon film, and allowing it to dry under ambient conditions.

#### 4.2.4. Z Potential

Zeta potential measurements were determined using the Zetasizer 2000 (Malvem Instruments Ltd., Worcestershire, UK). The voltage applied to drive the electrodes of the zeta cell was 150 V capillary electrophoresis. The sample was prepared by injecting approximately 2 mL solution into the cell.

### 4.3. Antimicrobial Activity

The bacterial strains used in this study were obtained from the stock culture collection of the Biochemistry laboratory at the School of Dentistry within the National Autonomous University of Mexico (UNAM). The strains were originally collected from clinical samples of the oral cavity of patients from the previously mentioned institution. These strains are native to central Mexico, having all been characterized through a set of culture media assays and biochemical tests. They included Gram-positive (*Staphylococcus aureus*, *Streptococcus mutans*, *Enterococcus faecalis*) and Gram-negative microorganisms (*Escherichia coli*) and yeast (*Candida albicans*), which are commonly present in the oral cavity and are responsible for important conditions in oral health. The experiments into antimicrobial activity were carried out as proscribed by the Clinical and Laboratory Standards Institute [69].

#### 4.3.1. Disk Diffusion Test

The NPs antibacterial properties were measured by the Kirby–Bauer disc diffusion method against the Gram-positive and Gram-negative bacteria and yeast. The inoculum was prepared by diluting the colonies with 0.9% of NaCl to 0.5 according to the McFarland scale, before they were applied to Muller–Hinton agar (Bioxon BD Mueller–Hinton II Agar) plates using sterile cotton swabs. The sterile paper discs were saturated with 10 μL of AgNPs (0.849 mg/mL) that had been prepared 24 h previously. In addition, three tubes with AgNPs that had been prepared 7, 14, and 21 days before the antimicrobial tests were used to saturate the paper discs and placed on agar plates. The disc impregnated with extract of clove was used as a positive control. After 24 and 48 h of incubation at 37 °C, the microbial susceptibility was determined through the measurement of any noticed inhibition zones. The assays were performed in triplicate [70].

The research protocol was reviewed and approved by the Ethics Committee of the Center for Research and Advanced Studies in Dentistry.

#### 4.3.2. Microdilution in Broth

The antimicrobial capacity of AgNPs were determined following the dilution method of the broth [71]. Selective media were used to grow each strain and then cultured on non-selective media. Samples were initially incubated at 37 °C for 24 h for fresh bacterial cultures, which were used to prepare McFarland standards. Then, 100 μL of Mueller–Hinton broth medium and positive control and a negative control (Mueller–Hinton broth and NPs as sterility control) were used. Each well was aseptically inoculated with 5 μL of the bacterial suspension (final concentration approximately 5 × 10^5^ CFU/mL) excluding controls. Subsequently, 100 μL of AgNPs (0.849 mg/mL) were placed at the beginning and seven serial microdilutions were made from the 100 μL of AgNPs. These assays were performed in triplicate in four wells for each concentration and strain. Inoculated microplates were incubated at 37 °C with continuous shaking at ~200 rpm for 24 h. The presence or absence of turbidity in each well was presented to the naked eye. The minimum concentration in the wells is taken as the MIC and the maximum effective concentration of MEC was identified by determining the lowest concentration of antimicrobial agent that reduces the viability of the initial bacterial inoculum by 99.9% or a log reduction in inoculum count [72].

## 5. Conclusions

AgNPs synthesized from aqueous extracts of *Sizygium aromaticum* show strong antibacterial activity. It is important to highlight that the nanoparticle generation method does not represent a strong environmental impact, it is very easy to carry out and, above all, very economical. In addition, this research has shown that AgNPs synthesized by this method are effective against the five most common microorganisms present in the oral cavity, these microorganisms are responsible for many oral health disorders, and we are sure that this research can contribute to the development of new antimicrobial agents for dental health or medical use, the routes of administration and cytotoxic assays will be exposed in future research because the expectations of this material are very high.

## Figures and Tables

**Figure 1 antibiotics-11-00834-f001:**
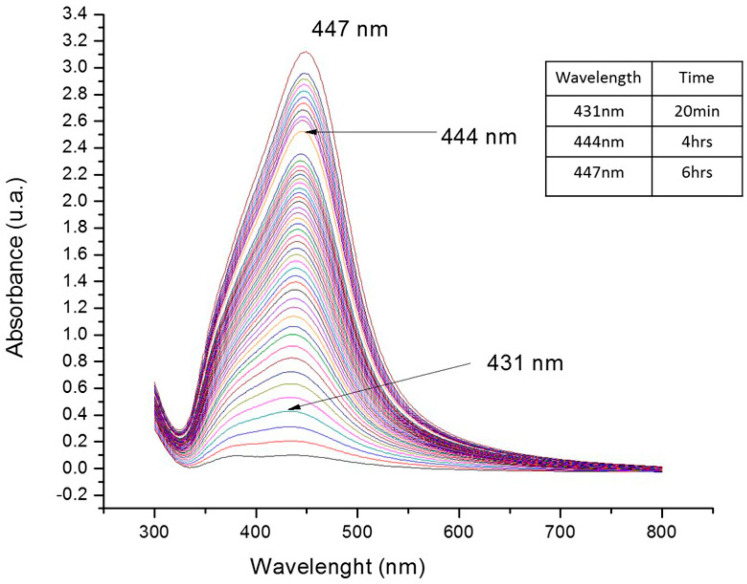
UV-Vis absorption spectrum of synthesized silver nanoparticles by *S. aromaticum* extract at different time intervals.

**Figure 2 antibiotics-11-00834-f002:**
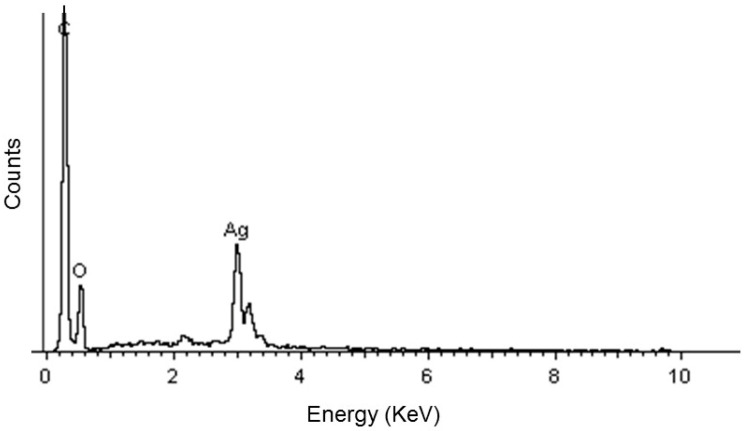
Scanning electron microscopy–energy dispersive X-ray spectrum of synthesized silver nanoparticles.

**Figure 3 antibiotics-11-00834-f003:**
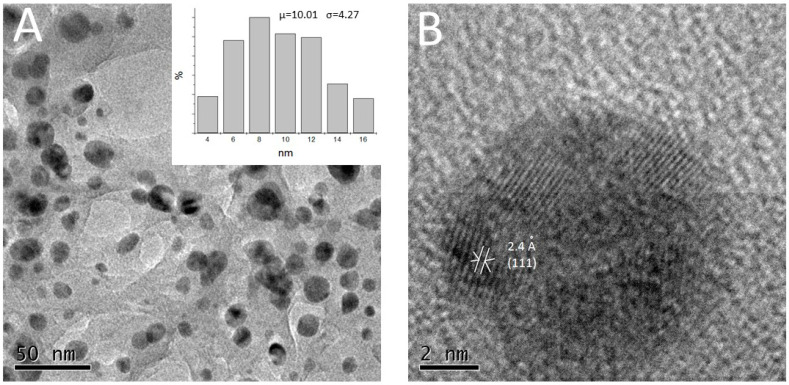
Characterization of synthesized biogenic silver nanoparticles from clove extract. (**A**) Micrograph of transmission electron microscopy, the insert shows size distribution and (**B**) high-resolution transmission electron microscopy micrograph.

**Figure 4 antibiotics-11-00834-f004:**
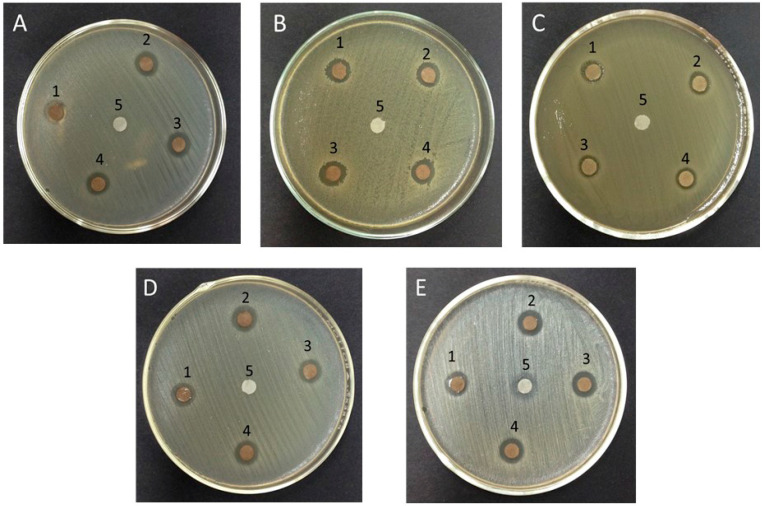
Antimicrobial activity against: (**A**) *Streptococcus mutans*, (**B**) *Staphylococcus aureus*, (**C**) *Escherichia coli*, (**D**) *Enterococcus faecalis*, (**E**) *Candida albicans*. Disk 1, 24 h after synthesis; disk 2, 7 days after synthesis; disk 3, 14 days after synthesis; disk 4, 21 days after synthesis; disk 5, control disk only with clove extract.

**Figure 5 antibiotics-11-00834-f005:**
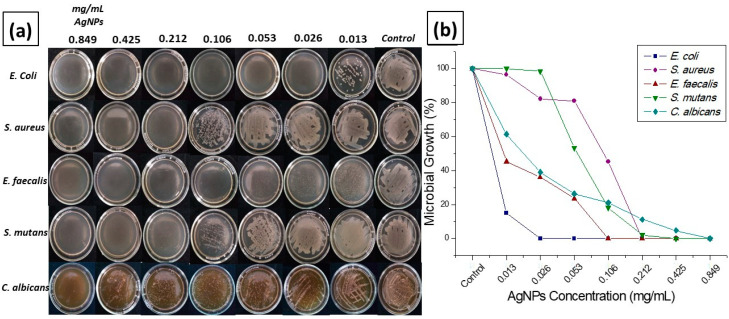
Microdilution in broth: (**a**) inhibition of microbial growth, (**b**) graph of antimicrobial growth against the applied concentration of nanoparticles.

**Table 1 antibiotics-11-00834-t001:** Minimum inhibitory concentration and maximum effective concentration of silver nanoparticles.

	AgNPs				
AgNPs (mg/mL)	*E. coli*	*S. aureus*	*E. faecalis*	*S. mutans*	*C. albicans*
0.849	-	-	-	-	-
0.425	-	-	-	-	+
0.212	-	-	-	+	+
0.106	-	+	-	+	+
0.053	-	+	+	+	+
0.026	-	+	+	+	+
0.013	+	+	+	+	+
Control	+	+	+	+	+

+ Microbial growth, - Without microbial growth MIC 
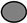
 MEC 
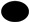
.

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
