# Peer review of "Antimicrobial Activity of Biogenic Silver Nanoparticles from Syzygium aromaticum against the Five Most Common Microorganisms in the Oral Cavity"

_antibiotics, 2022, doi:10.3390/antibiotics11070834_

Round 1

Reviewer 1 Report

The manuscript demonstrates the efficacy of silver nanoparticles synthesised from clove extract. It is an interesting research. However, there are several details that should be addressed. Please note the following comments:

General comments:

The nanoparticles used for this research were synthesised from a clove extract. Even though eugenol will be present on that extract, you cannot state that these are silver-eugenol nanoparticles since you did not use purified components of that extract nor did you use eugenol alone. Plus, why do you use the word synergic? Was there any synergetic activity detected? According to your methods and results, this does not seem to be the case. Please revise the title.

Regarding the relevance of this research for dentistry, this should be better contextualised and discussed.

Nanotechnological research has had many recent developments and you’re not taking advantage of any of this research. The majority of your references are not recent. You have no reference from this current year, only two from 2022, other two from 2020, and most of them older than this. I suggest checking the recent literature to better contextualise and discuss your data and results.

Be consistent, use the same wording for the same things; and, when using an acronym, make sure that you write it in full the first time, and from then on keep using the acronym.

Detailed comments and suggestions:

Line 25: Note that Gram should be written with capital letter. I suggest revising the entire document to correct this.

Line 27: You have not written the meaning for HR, just for TEM. Please revise.

Line 41: When you write “… unusually strong physicochemical and biological properties activities...”, I suggest writing “… unusually strong physicochemical properties and biological activities…”.

Line 42: Where you wrote “disinfecting medical devices”, do you mean “disinfectants, medical devices, and…”? If so, please revise and re-write.

Line 43: Please add a comma after the words “home appliances”.

Line 47: What is the method of preparation you’re referring to here? Chemical, physical or biological? Please state it clearly. Also, note that the main text of the manuscript is independent from the abstract. Therefore, any acronyms already written in full in the abstract need to be written in full again, for the first time they appear in the main text. Please write AgNPs in full and the acronym inside parenthesis.

Line 54: There are many other biological materials used on the production of metal nanoparticles, besides microorganisms, plants and their extracts. Please revise and complete this information.

Line 55: Even though many authors consider the biological processes as being non-toxic and environmentally friendly, these methods still make use of toxic chemical precursors like AgNO3. Furthermore, the NPs on their own can also have a negative environmental impact as stated in several recent studies. You might want to revise the text to “less toxic and more environmentally friendly than other methods”.

Line 63 and 66: You started by referring antimicrobial activity, but then you just write about bacteria. What about other organisms?

Line 69: Please detail on the “problems in the DNA’s replication”.

Lines 71-72: Please add a reference for this sentence.

Line 76: For each of the examples, please add a reference.

Line 77: Explain what is the controversy. Note that that there are clear proves of AgNPs’ impact on the environment and on different organisms.

Lines 86-88: This sentence reads more as you own opinion, but then you add 3 references. Is this your opinion supported by other authors? If so, clearly state this.

Line 93: The acronym “(UV-Vis)” should appear written right after “ultraviolet-visible; and, EDS should also be written in full.

Line 94: Note that zeta potential is not written with “Z”, but instead, it is written with a zeta (ζ). Revise the entire manuscript regarding this.

Line 107: Please write SPR’s in full first and then put the acronym within parenthesis.

Line 111: When you say that the NPs increase in size, do you mean that they aggregate?

Line 116: Please replace “Metallic silver nanoparticles” with AgNPs, and make sure you always use this acronym when referring to these from now on.

Lines 125-126: Note that legends of figures and tables are self-explanatory, in order to be understood independently from the main text. All acronyms used should be explained, either written in full or added in full as a footnote.

Line 134: Please replace the word “microorganisms” with “bacteria”.

Line 136: You say that the control group did not show any antimicrobial effect, however, in figure 4E, there is an hallo around the disc5.  Did you use the wrong figure or is this information not correct? Are you referring only to antibacterial activity not antifungal? Please revise.

Line 138: Note that the images show antimicrobial activity, not just antibacterial,

Line 140: Where it reads “(only clove)”, please write “(only clove extract)”.

Lines 146-148: for figure 5, I suggest increasing the size of both images a) and b). Please write the species names from figure 5b) in italics and consider changing the yy axis label from “bacterial/fungal growth” to “microbial growth”.

Line 151: Please note the previous comment from line 116.

Line 153: Please note the previous comment for lines 125-126.

Line 164: If the antifungal activity of the extract was compared to nystatin, was that activity higher or smaller? Please specify.

Line 173: You are now saying that the particles maintained their stability and uniformity, but previously you said that they increased in size. This is not the same thing. Please revise.

Line 174: Please add the word “extract” after the species name.

Lines 173 and 177: Please not the comment for line 116.

Line 187: Please correct the word “reaction”.

Line 205: What does FCC stand for? Is this acronym used anywhere else? If not consider removing it.

Line 206: What is “(111)”? Is this supposed to be a reference?

Lines 211-217: This paragraph is better suited if merged in the introduction. This way you will avoid unnecessarily repeating of information and the information will be better contextualized.

Lines 233 and 235: Please make sure that all species names are written in italics.

Line 238: Please remove the word “it” from the sentence “…it is low but not null.”.

Line 248: Please replace the word “salt” with “chemical precursor”. Use the chemical formula as you would use any acronym, write it within parenthesis and from now on use the chemical formula instead of the full name.

Line 251: the concentrations you are mentioning are form extract or from AgNO3? Please clarify.

Line 255: Does this deionized water had salt added or not? Please revise and change if necessary.

Line 257: what is the bio-reductor? What were the varying proportions? Please specify.

Line 262: What was this reducing agent? Please specify.

Line 272: what is this final product? Please be more specific.

Line 277: The full name of EDS should appear at its first mention, not here.

Line 280: What was this “simple suspension”?

Line 286: What cell are you referring to? Please clarify.

Lines 290-291: are the strains from clinical samples? Were these isolated from oral cavities? Why were these strains selected?

Line 298: Please replace the word “microorganisms” with “bacteria”.

Line 311: Where you wrote “…Ag the nanoparticles…” do you mean “AgNPs”? If so, please revise.

Line 312: For the “dilution method of the broth”, please either add a reference or explain it briefly.

Lines 311-325: Please revise the language in these sentences. English needs to be carefully revised here to make the sentences clear.

Line 316: What do you mean with “steric broth”?

Line 325: Note that, so far, you’ve used a different referencing system. This is not consistent with the rest of the manuscript.

Author Response

Responses to reviewer 1 observations:

The nanoparticles used for this research were synthesised from a clove extract. Even though eugenol will be present on that extract, you cannot state that these are silver-eugenol nanoparticles since you did not use purified components of that extract nor did you use eugenol alone. Plus, why do you use the word synergic? Was there any synergetic activity detected? According to your methods and results, this does not seem to be the case. Please revise the title.

We agree with reviewer 1, in fact there are other compounds present in the extract; however, it is reported in the literature that eugenol has excellent antibacterial properties; therefore, we infer that it is perhaps the component in the extract that possesses the main antibacterial activity, which, together with the excellent antimicrobial properties of the Ag nanoparticles, will enhance this activity. Nonetheless, the title has already been changed to avoid confusion and guesswork, we have marked the corrections in yellow color.

Regarding the relevance of this research for dentistry, this should be better contextualised and discussed.

We have contextualized and discussed the relevance to dentistry in the text.

Nanotechnological research has had many recent developments and you’re not taking advantage of any of this research. The majority of your references are not recent. You have no reference from this current year, only two from 2022, other two from 2020, and most of them older than this. I suggest checking the recent literature to better contextualise and discuss your data and results.

We agree with reviewer 1, the references have been updated, and the research importance in the nanotechnological field related to antibacterial activity has been recognized.

Be consistent, use the same wording for the same things; and, when using an acronym, make sure that you write it in full the first time, and from then on keep using the acronym.

The manuscript has been revised and homogenized as suggested by reviewer 1.

Detailed comments and suggestions:

Line 25: Note that Gram should be written with capital letter. I suggest revising the entire document to correct this.

The entire document has been modified and corrected; the changes are shown in yellow.

Line 27: You have not written the meaning for HR, just for TEM. Please revise.

The observation has been considered on line 27.

Line 41: When you write “… unusually strong physicochemical and biological properties activities...”, I suggest writing “… unusually strong physicochemical properties and biological activities…”.

It has been modified as suggested by the reviewer:

Recently, there has been renewed interest in manufactured silver nanomaterials, thanks to their unusually strong physicochemical properties and biological activities compared to their bulk parent materials [1,2].

Line 42: Where you wrote “disinfecting medical devices”, do you mean “disinfectants, medical devices, and…”? If so, please revise and re-write.

The sentence has been rewritten and corrected:

In addition, AgNPs synthesized by green methods have many other applications in different areas such as biotechnology, such as water filtration agents [3], disinfection and preservation of foods [4,5] and various materials, the production of cosmetics [5,6], nanoinsecticides and nanopesticides [7], nanocomposites [8], etc [9-14].

Line 43: Please add a comma after the words “home appliances”.

The entire sentence has been corrected and the word "home appliances" no longer appears.

Line 47: What is the method of preparation you’re referring to here? Chemical, physical or biological? Please state it clearly. Also, note that the main text of the manuscript is independent from the abstract. Therefore, any acronyms already written in full in the abstract need to be written in full again, for the first time they appear in the main text. Please write AgNPs in full and the acronym inside parenthesis.

We have indicated the method in the investigation in line 50:

Biological methods are extremely important, since reducing agents of a chemical nature are not required, but rather these are acquired from natural extracts and their compounds present in the aqueous extract, as is the case of the synthesis that is being used in This research also has great advantages because it is an easy and low-cost method [15].

The acronyms have been entered in the introduction with their corresponding abbreviation.

Line 43:

In addition, silver nanoparticles (AgNPs) synthesized by green methods have many other applications in different areas such as biotechnology, such as water filtration agents

Line 86:

Syzygium aromaticum (Clove) is employed in Indian ayurvedic medicine, Chinese medicine, and western herbalism, while in dentistry, its essential oil is used as an anodyne (painkiller) for emergencies.

Line 54: There are many other biological materials used on the production of metal nanoparticles, besides microorganisms, plants and their extracts. Please revise and complete this information.

The idea and the meaning of it has been rewritten:

reducing agents at the industrial level and within chemical methods are sodium borohydride, hydrazine and hypophosphite, which can increase environmental toxicity or biological hazards.

Line 55: Even though many authors consider the biological processes as being non-toxic and environmentally friendly, these methods still make use of toxic chemical precursors like AgNO3. Furthermore, the NPs on their own can also have a negative environmental impact as stated in several recent studies. You might want to revise the text to “less toxic and more environmentally friendly than other methods”.

We corrected the statement that we are incorrectly describing in the line 54.

Compared to chemical methods, biological methods represent less toxicity and are more respectful of the environment.

Line 63 and 66: You started by referring antimicrobial activity, but then you just write about bacteria. What about other organisms?

AgNPs show efficient antimicrobial properties compared to other metallic NPs, due to their large surface area, which provides better contact with microorganisms, although silver nanoparticles have been reported to be involved in a wide range of molecular processes within. of microorganisms, the mechanism of action is still being studied [16], it is important to note that there are not only bacteria that cause conditions in the oral cavity, but also some yeasts such as Candida albicans [17], recently it has been shown that AgNPs induce alterations in fungal cells and the formation of pores on the cell surface, in addition to changes in membrane fluidity and these may be related to changes in the lipid constitution of the plasma membrane and membrane depolarization [18]. Regarding the mechanism of action of AgNPs on bacteria, it has been studied that they have the ability to anchor and subsequently penetrate the bacterial cell wall, which causes structural changes and cell death [19].

Line 69: Please detail on the “problems in the DNA’s replication”.

DNA replication problems have been specified in line 85:

can lead to problems in the DNA’s replication of bacteria (such as epigenetic changes) [20]. Moreover, when it comes to microbial flora, the oral cavity is one of the most densely populated sites of the human body.

Lines 71-72: Please add a reference for this sentence.

has been properly referenced in line 82:

which electron spin resonance spectroscopy was used, suggest that free radicals form when the NPs come into contact with the bacteria [21]; these radicals are able to damage

Line 76: For each of the examples, please add a reference.

has been properly referenced in line 87:

Moreover, when it comes to microbial flora, the oral cavity is one of the most densely populated sites of the human body [22]. Over 700 bacterial species or phylotypes have been detected in this location, of which more than half have not been cultivated [23].

Line 77: Explain what is the controversy. Note that that there are clear proves of AgNPs’ impact on the environment and on different organisms.

Line 94: Despite the effectiveness shown by AgNPs in dental practice, controversy remains over their toxicity in biological and ecological systems, due to the cytotoxicity caused by high concentrations or the specific size of the nanoparticles. [24,25].

Lines 86-88: This sentence reads more as you own opinion, but then you add 3 references. Is this your opinion supported by other authors? If so, clearly state this.

It has been corrected, as it was an opinion and a goal to be achieved in future research:

Therefore, it is important to develop dental materials with antibacterial activity and superior mechanical properties, which could be manufactured and employed in future clinical applications.

Line 93: The acronym “(UV-Vis)” should appear written right after “ultraviolet-visible; and, EDS should also be written in full.

ultraviolet-visible (UV-Vis) spectrometry, scanning electron microscopy- energy dispersive x-ray spectroscopy (SEM-EDS).

Line 94: Note that zeta potential is not written with “Z”, but instead, it is written with a zeta (ζ). Revise the entire manuscript regarding this.

Abstract:

TEM (Transmission Electron Microscopy), and ζ Potential

Line 111: electron microscopy (TEM) and ζ Potential. The antibacterial activity of the AgNPs against

Line 107: Please write SPR’s in full first and then put the acronym within parenthesis.

Line 125: Each surface plasmon resonance (SPR) maximum was ubicated in the range of 431-447 nm.

Line 111: When you say that the NPs increase in size, do you mean that they aggregate?

were of small size and similar shape. As time increases, AgNPs stabilize and reach their maximum growth.

Line 116: Please replace “Metallic silver nanoparticles” with AgNPs, and make sure you always use this acronym when referring to these from now on.

We replaced in line 134: The characterization technique was consistent with the EDS analysis. AgNPs generally showed absorption peaks at approximately 3keV (Figure 2).

Lines 125-126: Note that legends of figures and tables are self-explanatory, in order to be understood independently from the main text. All acronyms used should be explained, either written in full or added in full as a footnote.

Figure 1. UV–Vis absorption spectrum of synthesized silver nanoparticles by S. aromaticum extract at different time intervals.

Figure 2. Scanning Electron Microscopy- Energy Dispersive x-ray spectrum of synthesized silver nanoparticles.

Figure 3. Characterization of silver nanoparticles with clove. (A) Micrograph of Transmission Electron Microscopy, the insert shows size distribution and (B) High-Resolution Transmission Electron Microscopy micrograph.

Table 1. Minimum inhibitory concentration and maximum effective concentration of silver nanoparticles.

Line 134: Please replace the word “microorganisms” with “bacteria”.

Line 153: The biogenic AgNPs were tested for antimicrobial activities against Gram-positive and Gram-negative bacteria and yeast.

Line 136: You say that the control group did not show any antimicrobial effect, however, in figure 4E, there is an hallo around the disc5.  Did you use the wrong figure or is this information not correct? Are you referring only to antibacterial activity not antifungal? Please revise.

Line 156: The control group (clove extract) did not show any antimicrobial effect only antifungal effect (Figure 4).

Line 138: Note that the images show antimicrobial activity, not just antibacterial,

Figure 4. Antimicrobial efficacy: (A) Streptococcus mutans, (B) Staphylococcus aureus, (C) Escherichia coli, (D) Enterococcus faecalis, (E) Candida albicans. Disk 1, 24 hrs, disk 2, 7

Line 140: Where it reads “(only clove)”, please write “(only clove extract)”.

Figure 4. Antimicrobial efficacy: (A) Streptococcus mutans, (B) Staphylococcus aureus, (C) Escherichia coli, (D) Enterococcus faecalis, (E) Candida albicans. Disk 1, 24 h, disk 2, 7 days after synthesis, disk 3. 14 days, disk 4. 21 days, disk 5, Control disk (only clove extract).

Lines 146-148: for figure 5, I suggest increasing the size of both images a) and b). Please write the species names from figure 5b) in italics and consider changing the yy axis label from “bacterial/fungal growth” to “microbial growth”.

Line 151: Please note the previous comment from line 116.

We have attended the comments throughout the document.

Line 153: Please note the previous comment for lines 125-126.

We have attended the comment in all figures.

Line 164: If the antifungal activity of the extract was compared to nystatin, was that activity higher or smaller? Please specify.

Line 186: C. albicans and compared it to nystatin. In both cases, the results obtained by the stratum being better than the antibiotic. The most probable pathway for AgNPs biosynthesis is

Line 173: You are now saying that the particles maintained their stability and uniformity, but previously you said that they increased in size. This is not the same thing. Please revise.

Line 111 has been revised and corrected, since we were referring to the nucleation and growth process of the nanoparticle and not to agglomerates.

Line 174: Please add the word “extract” after the species name.

The biosynthesis of AgNPs by clove extract is carried out rapidly, the solution changes color as a function of time, the kinetics of formation (fig.1) indicates that the reaction stabilizes after 6 hours.

Lines 173 and 177: Please not the comment for line 116.

We have attended the comments throughout the document.

Line 187: Please correct the word “reaction”.

After six hours of reaction, SPR with a defined shape appears at 447 nm, indicating that the NPs now have a defined shape and size, compared to plasmons at 431 nm at short reaction times, indicating incipient nanoparticle formation.

Line 205: What does FCC stand for? Is this acronym used anywhere else? If not consider removing it.

The interplanar distances and their corresponding crystalline planes match those of metallic Ag (Face-centered cubic structure).

Line 206: What is “(111)”? Is this supposed to be a reference?

This number between parenthesis corresponds to one of the main AgNPs crystal planes:

The measured interplanar distance was 2.4 Å, which corresponds to the plane {111}.

Lines 211-217: This paragraph is better suited if merged in the introduction. This way you will avoid unnecessarily repeating of information and the information will be better contextualized.

The paragraph has been removed at line interval 94-98

Lines 233 and 235: Please make sure that all species names are written in italics.

AgNPs as shown in Figure 5b, taking into account the efficacy in Escherichia coli having microbial effect until the last concentration of NPs. On the other hand, we observed bacterial growth of the Streptococcus mutans strain from the third highest concentration, also the following Staphylococcus aureus and Enterococcus faecalis, on the other hand, we see that the antifungal effectiveness of the AgNPs against Candida albicans it is low but not null.

Line 238: Please remove the word “it” from the sentence “…it is low but not null.”.

Line 259: the antifungal effectiveness of the AgNPs against Candida albicans is low but not null

Line 248: Please replace the word “salt” with “chemical precursor”. Use the chemical formula as you would use any acronym, write it within parenthesis and from now on use the chemical formula instead of the full name.

In this experimental study, the chemical precursor used was silver nitrate (AgNO3) (Sigma-Aldrich, St. Louis, MO, and EE.UU). The AgNPs were directly synthesized with the clove extact (Terana, Especias selectas, Ciudad de México, México) extract using a simple green synthesis procedure.

Line 251: the concentrations you are mentioning are form extract or from AgNO3? Please clarify.

A 10 mM solution of AgNO3 concentration was prepared using deionized water; To prepare the reducing agent, 1 g of chopped clove was added to 100 ml of boiling deionized water for five minutes.

Line 255: Does this deionized water had salt added or not? Please revise and change if necessary.

A 10 mM solution of AgNO3 concentration was prepared using deionized water; To prepare the reducing agent, 1 g of chopped clove was added to 100 ml of boiling deionized water for five minutes.

Line 257: what is the bio-reductor? What were the varying proportions? Please specify.

The mixture was allowed to cool before being filtered into a vacuum flask using a Buchner funnel and Whatman filter paper no. 5. The aqueous extract was mixed, at room temperature, with AgNO3 in 3:1 rate.

Line 262: What was this reducing agent? Please specify.

A 10 mM solution of AgNO3 concentration was prepared using deionized water; To prepare the reducing agent, 1 g of chopped clove was added to 100 ml of boiling deionized water for five minutes.

Line 272: what is this final product? Please be more specific.

The resultant mixture was kept undisturbed in a dark place. After a couple of hours, the color of the solution changed due to the formation of AgNPs. The process of biosynthesis was carried out under ambient environmental conditions (that is, at room temperature and under atmospheric pressure); the reaction was completed within a few minutes.

Line 277: The full name of EDS should appear at its first mention, not here.

We have made the corresponding changes and put the acronym correctly:

SEM-EDS

The final product was sonicated for 30 minutes to break up larger nanoparticle agglomerates; then, the particles were dried in a vacuum at room temperature (20 °C) prior to analysis. The NPs were attached to aluminum stubs with conductive tape, coated with carbon, and observed under SEM (JEOL, JSM-6510LV at 20 kcV, Tokyo, Japan) with secondary electrons at ×100, ×500, and ×3,000 magnification that was operating at 20 kV. EDS analysis was developed.

Line 280: What was this “simple suspension”?

We have corrected the wording:

TEM was obtained via a JEOL JEM-2100 microscope (Tokyo, Japan). Samples for the TEM examination were prepared by placing a drop of the suspension on a copper grid (300 mesh) coated with carbon film, and allowing it to dry under ambient conditions.

Line 286: What cell are you referring to? Please clarify.

The voltage applied to drive the electrodes of the zeta cell was 150V capillary electrophoresis.

Lines 290-291: are the strains from clinical samples? Were these isolated from oral cavities? Why were these strains selected?

Line: 309: The strains were originally collected from clinical samples of the oral cavity of patients from the same faculty, these samples are native to central Mexico; each one was characterized via a battery of cultural and biochemical tests. They included Gram-positive (Staphylococcus aureus, Streptococcus mutans, Enterococcus faecalis) and Gram-negative microorganisms (Escherichia coli), and yeast (Candida albicans), which are commonly present in the oral cavity and are responsible for important conditions in oral health.

Line 298: Please replace the word “microorganisms” with “bacteria”.

Line 318: The NPs antibacterial properties were measured by the Kirby-Bauer disc diffusion method against the Gram-positive and Gram-negative bacteria and yeast.

Line 311: Where you wrote “…Ag the nanoparticles…” do you mean “AgNPs”? If so, please revise.

Line 331: The antimicrobial capacity of AgNPs were determined following the dilution method of the broth [71].

Line 312: For the “dilution method of the broth”, please either add a reference or explain it briefly.

We have referenced the method in line 332:

The antimicrobial capacity of AgNPs were determined following the dilution method of the broth [71].

Lines 311-325: Please revise the language in these sentences. English needs to be carefully revised here to make the sentences clear.

The English wording has been thoroughly revised and aspects that were not clear in lines 331-344 have been changed.

Line 316: What do you mean with “steric broth”?

We refer to broth only without culture:

A positive control and a negative control (Mueller-Hinton broth and NPs as sterility control) were used.

Line 325: Note that, so far, you’ve used a different referencing system. This is not consistent with the rest of the manuscript.

We have put the reference according to the format of the journal:

Line 344: The minimum concentration in the wells is taken as the MIC and the maximum effective concentration of MEC is identified by determining the lowest concentration of antimicrobial agent that reduces the viability of the initial bacterial inoculum by 99.9% or a log reduction in inoculum count[72].

Our research group appreciates each and every one of the corrections and comments that you kindly sent us. We appreciate the time you spent improving this article.

References:

  1. Chaloupka, K.; Malam, Y.; Seifalian, A.M. Nanosilver as a new generation of nanoproduct in biomedical applications. Trends in biotechnology 2010, 28, 580-588.
  2. Gomaa, E.Z. Silver nanoparticles as an antimicrobial agent: A case study on Staphylococcus aureus and Escherichia coli as models for Gram-positive and Gram-negative bacteria. The Journal of general and applied microbiology 2017, 63, 36-43.
  3. Sanchooli, N.; Sanchooli, E.; Khandan Barani, H. Investigating the effect of water filter made using polyurethane foam containing silver nanoparticles on controlling Yersinia ruckeri in Oncorhynchus. Iranian Journal of Fisheries Sciences 2022, 21, 500-515.
  4. Liu, J.; Ma, Z.; Liu, Y.; Zheng, X.; Pei, Y.; Tang, K. Soluble soybean polysaccharide films containing in-situ generated silver nanoparticles for antibacterial food packaging applications. Food Packaging and Shelf Life 2022, 31, 100800.
  5. Pardo, L.; Arias, J.; Molleda, P. Elaboración de nanopartículas de plata sintetizadas a partir de extracto de hojas de romero (Rosmarinus officinalis L.) y su uso como conservante. LA GRANJA. Revista de Ciencias de la Vida 2022, 35, 45-58.
  6. Ong, W.T.J.; Nyam, K.L. Evaluation of silver nanoparticles in cosmeceutical and potential biosafety complications. Saudi Journal of Biological Sciences 2022.
  7. Awad, M.A.; Eid, A.M.; Elsheikh, T.M.Y.; Al-Faifi, Z.E.; Saad, N.; Sultan, M.H.; Selim, S.; Al-Khalaf, A.A.; Fouda, A. Mycosynthesis, Characterization, and Mosquitocidal Activity of Silver Nanoparticles Fabricated by Aspergillus niger Strain. Journal of Fungi 2022, 8, 396.
  8. Demchenko, V.; Kobylinskyi, S.; Iurzhenko, M.; Riabov, S.; Vashchuk, A.; Rybalchenko, N.; Zahorodnia, S.; Naumenko, K.; Demchenko, O.; Adamus, G. Nanocomposites based on polylactide and silver nanoparticles and their antimicrobial and antiviral applications. Reactive and Functional Polymers 2022, 170, 105096.
  9. Iravani, S. Green synthesis of metal nanoparticles using plants. Green Chemistry 2011, 13, 2638-2650.
  10. Amooaghaie, R.; Saeri, M.R.; Azizi, M. Synthesis, characterization and biocompatibility of silver nanoparticles synthesized from Nigella sativa leaf extract in comparison with chemical silver nanoparticles. Ecotoxicology and environmental safety 2015, 120, 400-408.
  11. Pani, A.; Lee, J.H.; Yun, S.-I. Autoclave mediated one-pot-one-minute synthesis of AgNPs and Au–Ag nanocomposite from Melia azedarach bark extract with antimicrobial activity against food pathogens. Chemistry Central Journal 2016, 10, 15.
  12. Jagtap, U.B.; Bapat, V.A. Green synthesis of silver nanoparticles using Artocarpus heterophyllus Lam. seed extract and its antibacterial activity. Industrial crops and products 2013, 46, 132-137.
  13. Huq, M.A.; Ashrafudoulla, M.; Rahman, M.M.; Balusamy, S.R.; Akter, S. Green synthesis and potential antibacterial applications of bioactive silver nanoparticles: A review. Polymers 2022, 14, 742.
  14. Salem, S.S.; Fouda, A. Green synthesis of metallic nanoparticles and their prospective biotechnological applications: an overview. Biological Trace Element Research 2021, 199, 344-370.
  15. González-Pedroza, M.G.; Argueta-Figueroa, L.; García-Contreras, R.; Jiménez-Martínez, Y.; Martínez-Martínez, E.; Navarro-Marchal, S.A.; Marchal, J.A.; Morales-Luckie, R.A.; Boulaiz, H. Silver nanoparticles from Annona muricata peel and leaf extracts as a potential potent, biocompatible and low cost antitumor tool. Nanomaterials 2021, 11, 1273.
  16. Ferreyra Maillard, A.P.V. Estudio del efecto antibacteriano de bio (nano) materiales contra microorganismos patógenos productores de mastitis bovina. 2019.
  17. Fonseca, M.S.; Rodrigues, D.M.; Sokolonski, A.R.; Stanisic, D.; Tomé, L.M.; Góes-Neto, A.; Azevedo, V.; Meyer, R.; Araújo, D.B.; Tasic, L. Activity of Fusarium oxysporum-Based Silver Nanoparticles on Candida spp. Oral Isolates. Nanomaterials 2022, 12, 501.
  18. Radhakrishnan, V.S.; Mudiam, M.K.R.; Kumar, M.; Dwivedi, S.P.; Singh, S.P.; Prasad, T. Silver nanoparticles induced alterations in multiple cellular targets, which are critical for drug susceptibilities and pathogenicity in fungal pathogen (Candida albicans). International journal of nanomedicine 2018, 13, 2647.
  19. Revati, S.; Bipin, C.; Chitra, P.B.; Minakshi, B. In vitro antibacterial activity of seven Indian spices against high level gentamicin resistant strains of enterococci. Archives of medical science: AMS 2015, 11, 863.
  20. Mandal, D.; Dash, S.K.; Das, B.; Chattopadhyay, S.; Ghosh, T.; Das, D.; Roy, S. Bio-fabricated silver nanoparticles preferentially targets Gram positive depending on cell surface charge. Biomedicine & Pharmacotherapy 2016, 83, 548-558.
  21. He, W.; Liu, Y.; Wamer, W.G.; Yin, J.-J. Electron spin resonance spectroscopy for the study of nanomaterial-mediated generation of reactive oxygen species. Journal of food and drug analysis 2014, 22, 49-63.
  22. Gao, L.; Xu, T.; Huang, G.; Jiang, S.; Gu, Y.; Chen, F. Oral microbiomes: more and more importance in oral cavity and whole body. Protein & cell 2018, 9, 488-500.
  23. Parahitiyawa, N.B.; Scully, C.; Leung, W.K.; Yam, W.C.; Jin, L.J.; Samaranayake, L.P. Exploring the oral bacterial flora: current status and future directions. Oral diseases 2010, 16, 136-145.
  24. García‐Contreras, R.; Argueta‐Figueroa, L.; Mejía‐Rubalcava, C.; Jiménez‐Martínez, R.; Cuevas‐Guajardo, S.; Sánchez‐Reyna, P.A.; Mendieta‐Zeron, H. Perspectives for the use of silver nanoparticles in dental practice. International dental journal 2011, 61, 297-301.
  25. Takahashi, N.; Nyvad, B. The role of bacteria in the caries process: ecological perspectives. Journal of dental research 2011, 90, 294-303.

Reviewer 2 Report

How the nanoparticles were dissolved for checking the antimicrobial activities ?

In the introduction section, importance of silver nanparticles is missing also the provide the various methodologies used for the nanoparticels with ins importance’s

In the abstract, characterization details of the prepared nanoparticles need to be included.

The Gram positive bacteria studied in this study is having oral cavity generation, if so kindly provide the evidence with references.

For the disc diffusion method provide the detailed composition of the media used and its preparation in the methodology section.

Also, elaborate how the prepared nanoparticles were dissolved?

Even though the characterization and antimicrobial activity of the nanoparticles were mentioned in the results section, the presentation of the final conclusion is very poor, especially the authors missed to include the novel outcome of the research work and future aspects of the work.

100 μl of the nanoparticles were used for the broth micro dilution technique, here the concertation of the nanoparticle need to be included. In addition, the positive control information and the concertation need to be presented.

The antimicrobial activity and how broth dilution values were determined are not clear for me and needs more data and explanation to support them.

Results and discussions are not clearly presented.  What authors trying to say in conclusion.

Author Response

Response to reviewer: 2

Comments and Suggestions for Authors

1- How the nanoparticles were dissolved for checking the antimicrobial activities ?

Comments:

Dear reviewer, before answering the questions and making the modifications that you kindly send us, the research group appreciates the time dedicated and the support for our research.

Regarding your first question, we have corrected the text because the nanoparticles were not really diluted at the beginning of the experiment, 100 μl of the initially obtained solution of AgNPs were placed and after adding to the first well they were diluted with the culture broth. This has been corrected in line 335 and marked green:

100 μl of Mueller-Hinton broth medium and 100 μl of AgNPs were placed in each well.

2- In the introduction section, importance of silver nanparticles is missing also the provide the various methodologies used for the nanoparticels with ins importance’s

 We agree with reviewer 2, since it was necessary to detail the importance of silver nanoparticles and also talk about the methodologies used for the generation of nanoparticles. We have given ourselves the task of further detailing this information, as you will see in the introduction, paragraphs have been added that we have carefully marked in yellow, enriching the investigation with the aforementioned details.

3- In the abstract, characterization details of the prepared nanoparticles need to be included.

 We have included details in the summary, both of the nanoparticle synthesis method, as well as of the characterization techniques, we mark these details in green:

Abstract: Syzygium aromaticum (clove) has been used as a dental analgesic, an anesthetic, and a bioreducing and capping agent in the formation of metallic nanoparticles. The main objective of this study was to evaluate the antimicrobial effect in oral microorganisms of biogenic silver nanoparticles (AgNPs) formed with aqueous extract of clove through an ecofriendly method “green synthesis”. The obtained AgNPs were characterized by UV-Vis (Ultraviolet-visible spectroscopy), SEM-EDS (Scanning Electron Microscopy- Energy Dispersive x-ray Spectroscopy), TEM (Transmission Electron Microscopy), and ζ Potential, while its antimicrobial effect was corroborated against oral Gram-positive and Gram-negative microorganisms, as well as yeast that is commonly present in the oral cavity. The AgNPs showed absorption at 400-500 nm in the UV-Vis spectrum, had an average size of 4-16 nm as observed by the High-Resolution Transmission Electron Microscopy (HR-TEM), and were of a crystalline nature and quasi-spherical form. The antimicrobial susceptibility test showed inhibition zones of 2-4 mm in diameter. Our results suggest that AgNPs synthesized with clove can be used as effective growth inhibitors in several oral microorganisms.

4-The Gram positive bacteria studied in this study is having oral cavity generation, if so kindly provide the evidence with references.

We have added text that can help reinforce why we are using Gram positive bacteria and that there is evidence that these strains and genera are present in the oral cavity. This information has been marked in green text and highlighted in purple.

[1]. In the normal oral cavity, several species of the genera Streptococcus, Lactobacillus, Lactococcus, Enterococcus, Staphylococcus, Corynebacterium, Veillonella and Bacteroides stand out, which are responsible for various oral conditions [2].

5-For the disc diffusion method provide the detailed composition of the media used and its preparation in the methodology section.

We have included details of the brand of the culture medium used, it has been marked in green text:

The inoculum was prepared by diluting the colonies with 0.9% of NaCl to 0.5 according to the McFarland scale, before they were applied to Muller Hinton agar (Bioxon BD Mueller Hinton II Agar) plates using sterile cotton swabs.

6-Also, elaborate how the prepared nanoparticles were dissolved?

 This observation has been resolved in comment number one.

7- Even though the characterization and antimicrobial activity of the nanoparticles were mentioned in the results section, the presentation of the final conclusion is very poor, especially the authors missed to include the novel outcome of the research work and future aspects of the work.

The conclusions have been rewritten and taken into account the valuable comments that helped to give relevance to the research:

 AgNPs synthesized from aqueous extracts of Sizygium aromaticum show strong antibacterial activity. It is important to highlight that the nanoparticle generation method does not represent a strong environmental impact, it is very easy to carry out and, above all, very economical. In addition, this research has shown that AgNPs synthesized by this method are effective against the five most common microorganisms present in the oral cavity, these microorganisms are responsible for many oral health disorders, and we are sure that this research can contribute to the development of new antimicrobial agents for dental health or medical use, the routes of administration and cytotoxic assays will be exposed in future research because the expectations of this material are very high.

8- 100 μl of the nanoparticles were used for the broth micro-dilution technique, here the concertation of the nanoparticle need to be included. In addition, the positive control information and the concertation need to be presented.

The information has been rewritten and details of the controls and the initial concentration in the 100 microliters have been given.

100 μl of Mueller-Hinton broth medium and positive control and a negative control (Mueller-Hinton broth and NPs as sterility control) were used. Each well is aseptically inoculated with 5 μl of the bacterial suspension (final concentration approximately 5x105 CFU/ml) excluding controls. Subsequently, 100 μl of AgNPs (0.849 mg/ml) were placed at the beginning and seven serial microdilutions were made from the 100 μl of AgNPs.

9- The antimicrobial activity and how broth dilution values were determined are not clear for me and needs more data and explanation to support them.

Information has been redrafted and described to help better understand the method:

The antimicrobial capacity of AgNPs were determined following the dilution method of the broth [3]. Where selective media were used to grow each strain and then cultured on non-selective media. Samples were initially incubated at 37°C for 24 h for fresh bacterial cultures, which were used to prepare McFarland standards. 100 μl of Mueller-Hinton broth medium and positive control and a negative control (Mueller-Hinton broth and NPs as sterility control) were used. Each well is aseptically inoculated with 5 μl of the bacterial suspension (final concentration approximately 5x105 CFU/ml) excluding controls. Subsequently, 100 μl of AgNPs (0.849 mg/ml) were placed at the beginning and seven serial microdilutions were made from the 100 μl of AgNPs. These assays were performed in triplicate in four wells for each concentration and strain. Inoculated microplates were incubated at 37 °C with continuous shaking at ~200 rpm for 24 h. The presence or absence of turbidity in each well is presented to the naked eye. The minimum concentration in the wells is taken as the MIC and the maximum effective concentration of MEC is identified by determining the lowest concentration of antimicrobial agent that reduces the viability of the initial bacterial inoculum by 99.9% or a log reduction in inoculum count[4].

10- Results and discussions are not clearly presented.  What authors trying to say in conclusion.

The results and discussions have been handled and worked in such a way that they agree with the conclusions set forth, the writing in the text has been improved and more details about the results set out in this research have been disclosed, as you will see in the new file that we send.

References:

  1. Harris, N.O.; Garcia-Godoy, F. Primary preventive dentistry; Upper Saddle River, NJ: Pearson Education: 2004.
  2. Alghamdi, S. Isolation and identification of the oral bacteria and their characterization for bacteriocin production in the oral cavity. Saudi journal of biological sciences 2022, 29, 318-323.
  3. Wayne, P. Reference method for broth dilution antifungal susceptibility testing of yeasts, approved standard. CLSI document M27-A2 2002.
  4. Kim, K.S.; Anthony, B.F. Importance of bacterial growth phase in determining minimal bactericidal concentrations of penicillin and methicillin. Antimicrobial agents and chemotherapy 1981, 19, 1075-1077.

Reviewer 3 Report

In Introduction, applications of nanomaterials in different fields should be marked, including regulations and safety aspects and related references should be added such as:

Souto et al. Nanopharmaceutics: Part I-Clinical Trials Legislation and Good Manufacturing Practices (GMP) of Nanotherapeutics in the EU. Pharmaceutics. 2020;12(2):146. Published 2020 Feb 11. doi:10.3390/pharmaceutics12020146

Yeung et la. 2020. Big impact of nanoparticles: analysis of the most cited nanopharmaceuticals and nanonutraceuticals research. Current Research in Biotechnology Volume 2, November 2020, Pages 53-63

Lines 89-96 should be rewritten.

The synthesis of Silver Nanoparticles  should be better described, including also a graphical scheme.

The description of results in "Section Results" should be implemented.

Results in Figura 4, Figure 5 and Table 1 should be better described in the text.

Limits, advantages, practical applications and future researches should be inserted in Conclusion.

Author Response

Response to reviewer: 3

Comments and Suggestions for Authors

In Introduction, applications of nanomaterials in different fields should be marked, including regulations and safety aspects and related references should be added such as:

Souto et al. Nanopharmaceutics: Part I-Clinical Trials Legislation and Good Manufacturing Practices (GMP) of Nanotherapeutics in the EU. Pharmaceutics. 2020;12(2):146. Published 2020 Feb 11. doi:10.3390/pharmaceutics12020146

We have inserted the valuable comments of reviewer 3 in the introduction, specifically on line 42, marked in blue.

Today it should be noted that nanomaterials have managed to enter different regulatory and safety fields (such as clinical trials and good manufacturing practices) [1], this to regulate and guarantee the quality of different areas of management and production [2].

Lines 89-96 should be rewritten.

these paragraphs have been rewritten again and marked in blue and yellow improved aspects, now this is found in lines 116-127.

Therefore, it is important to develop dental materials with antibacterial activity and better mechanical properties, which could be manufactured and employed in future clinical applications.

Herein, AgNPs were synthesized using aqueous clove extract, as a reducing and stabilizing agent, taking advantage of the AgNPs antimicrobial properties in synergy with the clove dental applications as an analgesic and anesthetic. The green-synthesized AgNPs were characterized by ultraviolet-visible (UV-Vis) spectrometry, scanning electron microscopy- energy dispersive x-ray spectroscopy (SEM-EDS), transmission electron microscopy (TEM) and ζ Potential. The antibacterial activity of the AgNPs against Gram-positive and Gram-negative microorganisms and yeast were tested by the disc-diffusion method.

The synthesis of Silver Nanoparticles  should be better described, including also a graphical scheme.

We have improved the wording of the nanoparticle generation process, it is a very simple method, therefore we better describe the proportions and specific details, we have marked the yellow color changes:

4.1. Synthesis of AgNPs

In this experimental study, the chemical precursor used was silver nitrate (AgNO3) (Sigma-Aldrich, St. Louis, MO, and EE.UU). The AgNPs were directly synthesized with the clove extact (Terana, Especias selectas, Ciudad de México, México) extract using a simple green synthesis procedure. Initially, various concentrations were tested until the optimum conditions were established.

A 10 mM solution of AgNO3 concentration was prepared using deionized water; to prepare the reducing agent, 1 g of chopped clove was added to 100 ml of boiling deionized water for five minutes. The mixture was allowed to cool before being filtered into a vacuum flask using a Buchner funnel and Whatman filter paper no. 5. The extract was mixed at room temperature with AgNO3 and bioreductant in 3:1 proportion. The resultant mixture was kept undisturbed in a dark place. After a couple of hours, the color of the solution changed due to the formation of AgNPs. The process of biosynthesis was carried out under ambient environmental conditions (that is, at room temperature and under atmospheric pressure); the reaction was completed within a few minutes.

The description of results in "Section Results" should be implemented.

The legend "section results" has been implemented in line 128 marked in blue

Results in Figura 4, Figure 5 and Table 1 should be better described in the text.

The indicated figures and table have been described in greater detail.

This last datum provides information regarding the efficacy provided by the clove extract only against Candida albicans, which turns out to be really important. In Figure 4E it is easy to perceive this important antifungal activity and it is observed to be slightly enhanced by the AgNPs. Regarding the results obtained in bacteria, as previously described, it is possible to see the activity of the potentiated extract.

Through this study we were able to determine both the maximum effective concentration (MEC), as well as the minimum inhibitory concentration (MIC), where we can observe the effectiveness of the NPs at different concentrations against the different microorganisms mentioned above (Figure 5a and Table 1). Specifically, in figure 5a it is possible to observe the complete inhibition at the highest concentration, likewise it is reflected in graph 5b how microbial growth is suddenly 100% inhibited by the effect of AgNPs (0.849 mg/ml). This information is reflected in Table 1. A very important result is that of the activity it has against the bacteria E. coli where the antibacterial activity is completely inhibited until the last treatment.

Limits, advantages, practical applications and future researches should be inserted in Conclusion.

The conclusions have been rewritten and taken into account the valuable comments that helped to give relevance to the research:

 AgNPs synthesized from aqueous extracts of Sizygium aromaticum show strong antibacterial activity. It is important to highlight that the nanoparticle generation method does not represent a strong environmental impact, it is very easy to carry out and, above all, very economical. In addition, this research has shown that AgNPs synthesized by this method are effective against the five most common microorganisms present in the oral cavity, these microorganisms are responsible for many oral health disorders, and we are sure that this research can contribute to the development of new antimicrobial agents for dental health or medical use, the routes of administration and cytotoxic assays will be exposed in future research because the expectations of this material are very high.

References:

  1. Souto, E.B.; Silva, G.F.; Dias-Ferreira, J.; Zielinska, A.; Ventura, F.; Durazzo, A.; Lucarini, M.; Novellino, E.; Santini, A. Nanopharmaceutics: Part I—Clinical trials legislation and good manufacturing practices (GMP) of nanotherapeutics in the EU. Pharmaceutics 2020, 12, 146.
  2. Yeung, A.W.K.; Souto, E.B.; Durazzo, A.; Lucarini, M.; Novellino, E.; Tewari, D.; Wang, D.; Atanasov, A.G.; Santini, A. Big impact of nanoparticles: Analysis of the most cited nanopharmaceuticals and nanonutraceuticals research. Current research in Biotechnology 2020, 2, 53-63.

Round 2

Reviewer 1 Report

The manuscript reads better now. However, there are still some minor issues that should be sorted. Please note the following comments:

Lines 2-4: I suggest the following for your title: “Antimicrobial activity of Biogenic Silver Nanoparticles from Syzygium aromaticum against the Five Most Common Microorganisms in the Oral Cavity”.

Lines 47-49: where you wrote: “different areas such as biotechnology, such as water filtration agents [10], disinfection and preservation of foods [11,12] and various materials, the production of cosmetics [12,13], nanoinsecticides and nanopesticides [14], nanocomposites [15], etc [7,16-20].”, I suggest writing “different biotechnological areas such as: development of water filtration agents [10]; disinfection and preservation of foods [11,12] and various materials [add reference here]; and, production of cosmetics [12,13], nanoinsecticides and nanopesticides [14], nanocomposites [15], amongst others (e.g., [7,16-20]).”.

Lines 53-57: The sentence is not clear. Revise and consider dividing into several smaller sentences to make the text more assertive.

Lines 71-80: Where you wrote “AgNPs show efficient antimicrobial properties compared to other metallic NPs, due to their large surface area, which provides better contact with microorganisms, although silver nanoparticles have been reported to be involved in a wide range of molecular processes within. of microorganisms, the mechanism of action is still being studied [32], it is important to note that there are not only bacteria that cause conditions in the oral cavity, but also some yeasts such as   [33], recently it has been shown that AgNPs induce alterations in fungal cells and the formation of pores on the cell surface, in addition to changes in membrane fluidity and these may be related to changes in the lipid constitution of the plasma membrane and membrane depolarization [34].”, I suggest revising and re-writing as follows: “AgNPs show efficient antimicrobial properties compared to other metallic NPs, due to their large surface area, which provides better contact with microorganisms. Although AgNPs have been reported to be involved in a wide range of molecular processes within microorganisms, the mechanism of action is still being studied [32]. It is important to note that there are not only bacteria that cause conditions in the oral cavity, but also some yeasts such as Candida albicans [33]. Recently, it has been shown that AgNPs induce alterations in fungal cells and the formation of pores on the cell surface, in addition to changes in membrane fluidity, all of which may be related to changes in the lipid constitution of the plasma membrane and membrane depolarization [34].”

Line 103: I suggest re-writing as follows: “… and Bacteroides stand out for being responsible for various oral conditions…”.

Line 108: Please delete the full stop before the references.

Line 128: Please delete the word “Section”.

Line 135: I suggest writing the title as “2.2. Characterization of synthesized silver nanoparticles”

Line 157: Where it reads: “Characterization of silver nanoparticles with clove.”, I suggest writing “Characterization of synthesized biogenic silver nanoparticles from clove extract”.

Line 176: I suggest writing “Figure 4. Antimicrobial activity against: (A) Streptococcus…”.

Lines 177-178: I suggest changing punctuation as follows: “Disk 1, 24 hours after synthesis; disk 2, 7 days after synthesis; disk 3, 14 days after synthesis; disk 4, 21 days after synthesis; and disk 5, Control disk only with clove extract.”

Lines 190-191: I suggest writing: “Microbial growth after exposure to determined concentrations of synthesized silver nanoparticles.”, and please change the name of the xx axis from “Concentration (mg/ml)” to “AgNPs Concentration (mg/ml)”.

Line 200: Please start this sentence as “Clove is one…”. Remove “it”.

Line 208: Note that stratum has a different meaning than extract, and sodium hypochlorite is not an antibiotic. Therefore, where you wrote “the results obtained by the stratum being better than the antibiotic.”, I suggest writing “the results obtained with the clove extracts were more effective than the substances they were compared to.”.

Line 226: Instead of “Accordingly to…”, please write “According to…”.

Line 228: Where you wrote “disponibility”, please replace with the word “availability”.

Line 287: Please remove the word “extact” written before the parenthesis.

Lines 290-292: Where you wrote “A 10 mM solution of AgNO3 concentration was prepared using deionized water; to prepare the reducing agent, 1 g of chopped clove was added to 100 ml of boiling deionized water for five minutes.”, I suggest writing “A 10 mM solution of AgNO3 was prepared using deionized water. To prepare the reducing agent, 1 g of chopped clove was added to 100 ml of boiling deionized water for five minutes.”.

Line 294: Please replace the word “rate” with “ratio” and make sure you specify the final volume prepared.

Line 299: I suggest changing the title as suggested for title 2.2..

Line 325: Instead of “faculty” I suggest writing “previously mentioned institution”.

Line 325-326: Where you wrote “…, these samples are native to central Mexico; each one was characterized via a battery of cultural and biochemical tests.”, I suggest writing this as a separate sentence, and re-writing it into something like: “These strains are native to central Mexico, having all been characterized through a set of culture media assays and biochemical tests.”.

Line 338: Please add the concentration of the AgNPs used here.

Line 342: Instead of “by the zones of inhibition.”, I suggest writing “… through the measurement of any noticed inhibition zones.”.

Line 343: Note that if the assays were performed in triplicate, you should have a basic statistical analysis. Having the standard deviations presented with values of diameter would be interesting, as well as doing a student t-test to demonstrate statistical significance of data.

Line 348: Remove the word “Where”, star the sentence with “Selective media were…”.

Line 352: Note that you are writing everything in the past tense. Please change “is” into “was”.

Lines 357-358: Where you wrote “The presence or absence of turbidity in each well is presented to the naked eye.”, I suggest you write “

Line 381: Make sure to write all scientific names, within your references, in italic.

Author Response

Thank you very much for the observations, the suggested changes have been made to each of the observations, these are marked in gray in the manuscript.

Lines 2-4: I suggest the following for your title: “Antimicrobial activity of Biogenic Silver Nanoparticles from Syzygium aromaticum against the Five Most Common Microorganisms in the Oral Cavity”.

Lines 47-49: where you wrote: “different areas such as biotechnology, such as water filtration agents [10], disinfection and preservation of foods [11,12] and various materials, the production of cosmetics [12,13], nanoinsecticides and nanopesticides [14], nanocomposites [15], etc [7,16-20].”, I suggest writing “different biotechnological areas such as: development of water filtration agents [10]; disinfection and preservation of foods [11,12] and various materials [add reference here]; and, production of cosmetics [12,13], nanoinsecticides and nanopesticides [14], nanocomposites [15], amongst others (e.g., [7,16-20]).”.

Lines 53-57: The sentence is not clear. Revise and consider dividing into several smaller sentences to make the text more assertive.

Lines 71-80: Where you wrote “AgNPs show efficient antimicrobial properties compared to other metallic NPs, due to their large surface area, which provides better contact with microorganisms, although silver nanoparticles have been reported to be involved in a wide range of molecular processes within. of microorganisms, the mechanism of action is still being studied [32], it is important to note that there are not only bacteria that cause conditions in the oral cavity, but also some yeasts such as   [33], recently it has been shown that AgNPs induce alterations in fungal cells and the formation of pores on the cell surface, in addition to changes in membrane fluidity and these may be related to changes in the lipid constitution of the plasma membrane and membrane depolarization [34].”, I suggest revising and re-writing as follows: “AgNPs show efficient antimicrobial properties compared to other metallic NPs, due to their large surface area, which provides better contact with microorganisms. Although AgNPs have been reported to be involved in a wide range of molecular processes within microorganisms, the mechanism of action is still being studied [32]. It is important to note that there are not only bacteria that cause conditions in the oral cavity, but also some yeasts such as Candida albicans [33]. Recently, it has been shown that AgNPs induce alterations in fungal cells and the formation of pores on the cell surface, in addition to changes in membrane fluidity, all of which may be related to changes in the lipid constitution of the plasma membrane and membrane depolarization [34].”

Line 103: I suggest re-writing as follows: “… and Bacteroides stand out for being responsible for various oral conditions…”.

Line 108: Please delete the full stop before the references.

Line 128: Please delete the word “Section”.

Line 135: I suggest writing the title as “2.2. Characterization of synthesized silver nanoparticles”

Line 157: Where it reads: “Characterization of silver nanoparticles with clove.”, I suggest writing “Characterization of synthesized biogenic silver nanoparticles from clove extract”.

Line 176: I suggest writing “Figure 4. Antimicrobial activity against: (A) Streptococcus…”.

Lines 177-178: I suggest changing punctuation as follows: “Disk 1, 24 hours after synthesis; disk 2, 7 days after synthesis; disk 3, 14 days after synthesis; disk 4, 21 days after synthesis; and disk 5, Control disk only with clove extract.”

Lines 190-191: I suggest writing: “Microbial growth after exposure to determined concentrations of synthesized silver nanoparticles.”, and please change the name of the xx axis from “Concentration (mg/ml)” to “AgNPs Concentration (mg/ml)”.

Line 200: Please start this sentence as “Clove is one…”. Remove “it”.

Line 208: Note that stratum has a different meaning than extract, and sodium hypochlorite is not an antibiotic. Therefore, where you wrote “the results obtained by the stratum being better than the antibiotic.”, I suggest writing “the results obtained with the clove extracts were more effective than the substances they were compared to.”.

Line 226: Instead of “Accordingly to…”, please write “According to…”.

Line 228: Where you wrote “disponibility”, please replace with the word “availability”.

Line 287: Please remove the word “extact” written before the parenthesis.

Lines 290-292: Where you wrote “A 10 mM solution of AgNO3 concentration was prepared using deionized water; to prepare the reducing agent, 1 g of chopped clove was added to 100 ml of boiling deionized water for five minutes.”, I suggest writing “A 10 mM solution of AgNO3 was prepared using deionized water. To prepare the reducing agent, 1 g of chopped clove was added to 100 ml of boiling deionized water for five minutes.”.

Line 294: Please replace the word “rate” with “ratio” and make sure you specify the final volume prepared.

Line 299: I suggest changing the title as suggested for title 2.2..

Line 325: Instead of “faculty” I suggest writing “previously mentioned institution”.

Line 325-326: Where you wrote “…, these samples are native to central Mexico; each one was characterized via a battery of cultural and biochemical tests.”, I suggest writing this as a separate sentence, and re-writing it into something like: “These strains are native to central Mexico, having all been characterized through a set of culture media assays and biochemical tests.”.

Line 338: Please add the concentration of the AgNPs used here.

Line 342: Instead of “by the zones of inhibition.”, I suggest writing “… through the measurement of any noticed inhibition zones.”.

Line 343: Note that if the assays were performed in triplicate, you should have a basic statistical analysis. Having the standard deviations presented with values of diameter would be interesting, as well as doing a student t-test to demonstrate statistical significance of data.

Line 348: Remove the word “Where”, star the sentence with “Selective media were…”.

Line 352: Note that you are writing everything in the past tense. Please change “is” into “was”.

Lines 357-358: Where you wrote “The presence or absence of turbidity in each well is presented to the naked eye.”, I suggest you write “

Line 381: Make sure to write all scientific names, within your references, in italic.

Reviewer 2 Report

The revised manuscript has been improved and it can be accepted in the present format

Author Response

Thank you very much for your contributions.
